# Multiple Observations for Secret-Key Binding with SRAM PUFs

**DOI:** 10.3390/e23050590

**Published:** 2021-05-11

**Authors:** Lieneke Kusters, Frans M. J. Willems

**Affiliations:** Information and Communication Theory Lab, Signal Processing Systems Group, Department of Electrical Engineering, Eindhoven University of Technology, 5600 MB Eindhoven, The Netherlands; f.m.j.willems@tue.nl

**Keywords:** secret-key agreement, Physical Unclonable Functions, helper data scheme, LDPC code

## Abstract

We present a new Multiple-Observations (MO) helper data scheme for secret-key binding to an SRAM-PUF. This MO scheme binds a single key to multiple enrollment observations of the SRAM-PUF. Performance is improved in comparison to classic schemes which generate helper data based on a single enrollment observation. The performance increase can be explained by the fact that the reliabilities of the different SRAM cells are modeled (implicitly) in the helper data. We prove that the scheme achieves secret-key capacity for any number of enrollment observations, and, therefore, it is optimal. We evaluate performance of the scheme using Monte Carlo simulations, where an off-the-shelf LDPC code is used to implement the linear error-correcting code. Another scheme that models the reliabilities of the SRAM cells is the so-called Soft-Decision (SD) helper data scheme. The SD scheme considers the one-probabilities of the SRAM cells as an input, which in practice are not observable. We present a new strategy for the SD scheme that considers the binary SRAM-PUF observations as an input instead and show that the new strategy is optimal and achieves the same reconstruction performance as the MO scheme. Finally, we present a variation on the MO helper data scheme that updates the helper data sequentially after each successful reconstruction of the key. As a result, the error-correcting performance of the scheme is improved over time.

## 1. Introduction

The The Internet of Things (IoT) makes it possible to connect and share information between many different devices through the Internet. This sharing of information is beneficial for many applications, e.g., in healthcare or consumer electronics. At the same time, the information may be sensitive and should not fall into the wrong hands or be tampered with. Therefore, security is one of the main challenges of the IoT devices. Since the IoT devices are often small and low cost, securing the devices should come at a low price.

Secure communication is often achieved through cryptographic protocols that rely on secret keys. A low-cost alternative for secure storage of the keys is enabled by Static Random-Access Memory Physical Unclonable Functions (SRAM PUFs). A PUF is a physical object or device that responds to a challenge with a response that is unique and unpredictable [1,2]. The SRAM-PUF functionality is based on the uninitialized values of the SRAM. These are the values that appear in the memory cells directly after power up of the SRAM. The corresponding binary vector is unique for each SRAM and it is the result of small variations in the silicon material. It can be considered to be a noisy fingerprint of the device and can be used to generate and bind secret keys [3,4].

Since the SRAM-PUF observations are noisy, additional processing is required to ensure reliable reconstruction of the key. This can be achieved through a so-called key binding scheme, see Figure 1. The scheme considers two phases: an enrollment phase during which a uniformly generated key is bound to a first SRAM-PUF observation; and a reconstruction phase during which the key is reconstructed from an additional observation of the SRAM-PUF. Please note that enrollment is usually performed only once, whereas reconstruction can be repeated many times. During enrollment, besides the key *s*, also some helper data *w* is generated. This helper data *w* ensures that the key can be reliably reconstructed even though $y^{n}$ is a noisy version of $x^{n}$. The helper data are considered public, and therefore should not reveal information about the key *s* to an attacker.

In classic helper data schemes, a single SRAM-PUF observation is used for enrollment and a single observation is used for reconstruction. However, it has been shown that the mutual information between the encoder and decoder observations increases when more observations are considered [5]. Since the secret-key capacity is equal to the mutual information (see Section 3), it follows that the achievable secret-key rate is increased when multiple SRAM-PUF observations are used instead of a single observation.

We introduce the Multiple-Observations (MO) helper data scheme, which enrolls a single key after processing multiple SRAM-PUF observation vectors. The scheme is based on a linear error-correcting code and can be seen as an extension of the fuzzy commitment scheme [6]. Any number of enrollment observations can be used. Furthermore, the performance of the scheme increases when more observations are used.

### 1.1. Related Work

A first implementation of a key binding scheme for generating and reconstructing a cryptographic key from SRAM PUFs was presented by Guajardo et al. in [4]. There, the fuzzy commitment scheme [6] was used to construct the helper data and later to reproduce the key. It is known that within one SRAM-PUF, some cells are more reliable (smaller error probability) than other cells [3]. The reliability information of the SRAM cells can be used to improve the performance of the helper data schemes. For example, a Soft-Decision (SD) helper data scheme [7,8] publicly shares the error probability of each SRAM cell to improve the decoder performance. Furthermore, a Selection-based helper data scheme [9,10,11] selects only the most reliable SRAM cells to reduce the average error probability of the SRAM-PUF observations. Both schemes assume that reliability of each SRAM cell is known during enrollment. However, in general this information is not available. Either, special measurement techniques must be applied, or a sufficient number of observations is required to estimate these values before enrollment. We propose a new scheme that accepts standard SRAM-PUF observation vectors as an input, i.e., the MO helper data scheme. In [12] multiple enrollment observations are used under various environmental conditions. The focus of [12] is on experimental validation of new strategies that consider multiple enrollment temperatures; however, a mathematical analysis of the strategies is missing. Our work focuses instead on finding an optimal multiple observations strategy. In the future, our analysis may be extended to consider temperature dependence as well, see Section 2.

Applying multiple observations for key binding (and generation) has been studied from information-theoretic perspective in [13,14,15]. It is shown that the secret-key rate can be improved when multiple observations are used by the encoder or the decoder. Achievable rate regions are analyzed for various multiple enrollments and multiple entities scenarios, but no code constructions are proposed or investigated.

Multiple enrollment scenarios are studied from leakage perspective in [16,17]. In these papers, scenarios are considered (e.g., the reverse fuzzy extractor [18]) in which enrollment is repeated multiple times and correspondingly multiple helper data are generated. Please note that the additional helper data are not generated to increase performance; instead, they are considered to be a security challenge that follows from the repeated enrollments. The decoder performs a classic reconstruction that is based on a single helper data sequence only, whereas an attacker may have stored all previously generated helper data. It is shown in [17] that zero leakage is ensured in these multiple enrollment scenarios, when the SRAM PUFs meet a certain symmetry condition. Inspired by the zero leakage results of [17], we have developed the MO helper data scheme which produces a single helper data sequence based on repeated enrollments. In contrast to the scenarios discussed in the previous paragraph, the decoder now benefits from the additional observations which are embedded in the helper data.

### 1.2. Contributions and Outline

In Section 3, we define secret-key capacity for multiple enrollment observations and calculate its value for the SRAM-PUF statistical model. We show that secret-key capacity can increase significantly when more enrollment observations are considered. This observation is the main motivation for the results presented in this paper, which are listed as follows.

We introduce the MO helper data scheme in Section 4. We prove that the helper data does not reveal any information about the key when the SRAM-PUF statistical model meets the symmetry assumption (Section 2). Then, we prove that secret-key capacity can be achieved with the MO helper data scheme for any number of enrollment observations.In Section 5, we present a code construction and evaluate performance of the scheme through Monte Carlo simulations.We propose a new variation on the Soft-Decision (SD) helper data scheme from [7] in Section 6. In contrast to the original scheme, this scheme considers binary SRAM-PUF observations as an input. We prove that the new SD strategy is optimal (achieves secret-key capacity) and that it can achieve the same reconstruction performance as the MO helper data scheme.In Section 7, we present a variation on the MO helper data scheme that can update the helper data sequentially. The error-correcting performance of the scheme improves after each successful key reconstruction, and therefore the performance improves over the lifetime of the device. This enables usage of less observations during the enrollment phase, when allowing worse initial reconstruction performance that is improved over time.

We conclude with a summary of our results. In the following, we first introduce the notation and the statistical model that we use for SRAM PUFs.

## 2. Notation and SRAM-PUF Statistical Model

In the following, we first introduce the notation that is used. Then, we present the statistical model that we use for SRAM PUFs which is based on the commonly used model introduced in [7]. We introduce a symmetry assumption which is required for the security of the MO helper data scheme. We derive several properties that are needed for the proofs later in this work.

### 2.1. Notation and Definitions

We use uppercase symbols to denote random variables and lowercase symbols to denote their realizations. We consider *t* enrollment observations of *n* SRAM cells, corresponding to *t* binary observation vectors of length *n*, i.e., $\left(x_{1}^{n},x_{2}^{n},\ldots ,x_{t}^{n}\right)$. The *i*th observation of the *j*th SRAM cell is represented by $x_{i,j}$, and assumes a value in {0,1}. We often analyze the behavior of a single SRAM cell, in which case we omit the cell index *j* and we have $\left(x_{1},x_{2},\ldots ,x_{t}\right)$, corresponding to *t* observations of one SRAM cell.

The reconstruction observation vector is represented as $y^{n}$, with $y_{j}$ the reconstruction observation of the *j*th SRAM cell. We use a distinct symbol for the reconstruction observation to emphasize its different functionality in the helper data scheme. Note, however, that the reconstruction observations have the same statistical behavior as the enrollment observations, and they may as well have been represented as $x_{t+1}^{n}$, the (t+1)th observation of the SRAM cells.

Calligraphic letters are used for finite sets and |S| denotes the cardinality of the set S. Finally, (conditional) entropy and mutual information are defined as in [19,20], with the base of the log equal to 2 and the units are bits. We define the binary entropy function as
(1)$$h_{2}\left(p\right)\overset{\Delta }{=}-p\log_{2}\left(p\right)-\left(1-p\right)\log_{2}\left(1-p\right).$$

### 2.2. Sram-Puf Statistical Model

#### 2.2.1. One-Probability of a Cell

Each SRAM cell has a one-probability θ∈[0,1] that defines the probability that a one is observed for this cell, i.e.,
(2)$$\mathrm{Pr}\left(X=1|\Theta =\theta \right)=\theta .$$
In practice, we do not know the one-probability θ of each SRAM cell. Instead, the one-probability is a random variable, independent and identically distributed over the cells according to some known distribution $p_{\Theta }\left(\theta \right)$. And we can calculate the average one-probability
(3)$$\mathrm{Pr}\left(X=1\right)=\int_{0}^{1}\theta p_{\Theta }\left(\theta \right)\;d\theta .$$

The currently most used model for PUFs [7] adopts the following Θ-distribution
(4)$$p_{\Theta }\left(\theta \right)=\frac{\lambda_{1}\phi \left(\lambda_{2}-\lambda_{1}\Phi^{-1}\left(\theta \right)\right)}{\phi \left(\Phi^{-1}\left(\theta \right)\right)}.$$
Here $\phi \left(x\right)$ and $\Phi^{-1}\left(x\right)$ are the probability density function and the inverse of the cumulative distribution function of the normal distribution. Furthermore, $\lambda_{1}$ and $\lambda_{2}$ are parameters that determine the average reliability and average one-probability of the SRAM cells, respectively.

It has been observed on several occasions [3,4,21] that the SRAM PUFs are *unbiased*, i.e.,
(5)Pr(X=1)=Pr(X=0)=1/2,
the average probability of observing a one is equal to the probability of observing a zero. This is a desirable property for PUFs as it ensures that the observations are completely unpredictable for an attacker. In [7], Maes et al. showed that the model (Equation 4) well represents their empirical data for $\lambda_{1}=0.065,\lambda_{2}=0.000$, which indeed corresponds to unbiased SRAM PUFs. In this paper, we assume an unbiased SRAM-PUF and set $\lambda_{2}=0$. It follows that the distribution of the one-probabilities is *symmetric*, i.e.,
(6)$$p_{\Theta }\left(\theta \right)=p_{\Theta }\left(1-\theta \right).$$
This symmetry of the one-probability distribution is a property of the SRAM-PUF model that is key for the zero-secrecy leakage of the multiple observation schemes discussed in this paper.

#### 2.2.2. Multiple Observations

First, we consider multiple observations of a single SRAM cell. The probability of observing *t* observations $\left(x_{1},x_{2},\ldots ,x_{t}\right)$ of an SRAM cell with Hamming weight $k=w_{H}\left(x_{1},x_{2},\ldots ,x_{t}\right)$ (where the Hamming weight is defined as the number of non-zero symbols in the sequence), is defined as
(7)$$\begin{array}{cc} \pi_{t}\left(k\right) & \overset{\Delta }{=}\mathrm{Pr}\left(X_{1}=x_{1},\ldots ,X_{t}=x_{t}\right)=\int_{0}^{1}\mathrm{Pr}\left(X_{1}=x_{1},\ldots ,X_{t}=x_{t}|\Theta =\theta \right)p_{\Theta }\left(\theta \right)\;d\theta \\ & =\int_{0}^{1}\theta^{k}{\left(1-\theta \right)}^{t-k}p_{\Theta }\left(\theta \right)\;d\theta \overset{\left(a\right)}{=}\int_{0}^{1}{\left(1-\theta^{\prime }\right)}^{k}{\left(\theta^{\prime }\right)}^{t-k}p_{\Theta }\left(\theta^{\prime }\right)\;d\theta^{\prime }=\pi_{t}\left(t-k\right), \end{array}$$
where in $\left(a\right)$ we substituted with $\theta^{\prime }=\left(1-\theta \right)$ and applied the symmetry property (Equation 6). Please note that the probability only depends on the number of observed ones *k*, and the *order of the observations is irrelevant*. Furthermore, the probability of observing a length *t* sequence containing *k* ones is equal to the probability of observing a length *t* sequence containing *k* zeros. Similarly, the conditional probability of observing a value *y* given a previously observed sequence $\left(x_{1},x_{2},\ldots ,x_{t}\right)$, only depends on the number of previously observed ones *k*, i.e.,
(8)$$\begin{array}{cc} \mathrm{Pr}\left(Y=y|X_{1}=x_{1},\ldots ,X_{t}=x_{t}\right) & \overset{\left(a\right)}{=}\frac{\mathrm{Pr}\left(Y=y,X_{1}=x_{1},\ldots ,X_{t}=x_{t}\right)}{\mathrm{Pr}\left(X_{1}=x_{1},\ldots ,X_{t}=x_{t}\right)}=\frac{\pi_{t+1}\left(k+y\right)}{\pi_{t}\left(k\right)}\\ & =\mathrm{Pr}\left(Y=y|w_{H}\left(X_{1},\ldots ,X_{t}\right)=k\right)\overset{\left(b\right)}{=}\frac{\pi_{t+1}\left(t+1-\left(k+y\right)\right)}{\pi_{t}\left(t-k\right)}\\ & =\mathrm{Pr}\left(Y=y⊕1|w_{H}\left(X_{1},\ldots ,X_{t}\right)=t-k\right), \end{array}$$
where in $\left(a\right)$ we used Bayes’ theorem and in $\left(b\right)$ we used (7). Finally, the last step follows from the fact that $\left(1-y\right)\equiv \left(y⊕1\right)$ for y∈{0,1}.

#### 2.2.3. Multiple SRAM Cells

All the equations that were derived for a single SRAM cell can be easily generalized to multiple SRAM cells. Since the one-probabilities are independent and identically distributed over the SRAM cells, the observations of different cells are also independent and the joint probability of *t* observations for *n* cells is
(9)$$\mathrm{Pr}\left(X_{1}^{n}=x_{1}^{n},X_{2}^{n}=x_{2}^{n},\ldots ,X_{t}^{n}=x_{t}^{n}\right)=\prod_{j=1}^{n}\mathrm{Pr}\left(X_{1}=x_{1,j},\ldots ,X_{t}=x_{t,j}\right)=\prod_{j=1}^{n}\pi_{t}\left(k_{j}\right),$$
with $k_{j}=w_{H}\left(x_{1,j},\ldots ,x_{t,j}\right)$ the number of observed ones after *t* observations of the *j*th cell.

#### 2.2.4. Θ-Distribution

Until now, we have assumed that $\lambda_{2}=0$ and thus the SRAM-PUF is unbiased. This assumption is sufficient for the security and achievability proofs in Section 4. However, to predict the performance of our scheme in a practical setting, we also need to set a value for $\lambda_{1}$. In correspondence with previous works [7,11] we choose $\lambda_{1}=0.51$ for our simulations, which corresponds to average error probability regarding dominant (most likely) value of a cell
(10)$$\bar{\psi }\overset{\Delta }{=}\int_{0}^{1}\min\left(\theta ,\left(1-\theta \right)\right)p_{\Theta }\left(\theta \right)\;d\theta \approx 0.15.$$

In Figure 2, we plot the Θ-distribution that is used for the calculations later in the paper. Please note that stable cells (one-probability close to 0 or 1) are more likely than unstable cells. We will see later that knowledge about the stability (error probability) of specific cells can improve the performance of helper data schemes.

Finally, note that temperature and voltage ramp-up time of the power up may influence the behavior of the SRAM PUFs [21]. Here, we do not model this behavior and only assume a worst-case average error probability of 0.15 (see (Equation 8)). However, it is worthwhile mentioning that the model (Equation 4) can be extended to also model the temperature dependence, see [22]. Knowledge about the temperature behavior can be exploited to improve the performance of the helper data schemes, especially when the temperature is known [23]. Therefore, in the future, it may be beneficial to extend the current work to also consider temperature dependent behavior of the SRAM PUFs.

## 3. Multiple Enrollment Observations for Increased Secret-Key Capacity

In this section, we describe a secret-key binding scheme with multiple enrollment observations, see Figure 3. We derive the achievable secret-key rate for the used statistical model and show that the rate can improve significantly when more enrollment observations are considered. These results motivate us to design a helper data scheme that can approach such rates in Section 4.

As in the classic (single enrollment) key binding scheme in Figure 1, we distinguish between an enrollment and a reconstruction phase.

During enrollment, the secret key s∈{1,2,…,|S|} is generated uniformly at random. An encoder obtains the secret key *s* and *t* observation vectors $\left(x_{1}^{n},x_{2}^{n},\ldots ,x_{t}^{n}\right)$, and generates corresponding helper data w∈W. The helper data alphabet W is specified by the encoding function that is used, see Section 4. We assume that the helper data *w* is stored and/or communicated in the public domain. Therefore, the helper data by itself should not reveal any information about the key. During reconstruction, a decoder observes a reconstruction vector $y^{n}$ as well as the helper data *w* and maps this input to an estimate $\hat{s}$ of the secret key. Reconstruction is successful when the estimate is equal to the original key, i.e., when $\hat{s}=s$.

We are now interested in the achievable secret-key rate of the scheme when enrollment is performed using *t* observation vectors of the SRAM-PUF, and where achievable rate is defined as follows.

**Definition** **1.**
*A secret-key rate $R_{t}$ is called achievable after t enrollment observations, if for all δ>0 and for all n large enough, there exist encoders and decoders such that*
(11)$$\mathrm{Pr}\left(\hat{S}\neq S\right)\leq \delta ,$$
(12)$$\frac{1}{n}H\left(S\right)=\frac{1}{n}\log_{2}\left|S\right|\geq R_{t}-\delta ,$$
(13)$$\frac{1}{n}I\left(S;W\right)\leq \delta .$$
*The secret-key capacity $C_{t}$ is the maximum achievable secret-key rate with t enrollment observations.*


Here (11) requires that the key reconstruction is reliable, (12) that the key is uniformly distributed, and (13) limits the information leakage by the helper data about the key. Theorem 1 gives the fundamental limit on achievable secret-key rate for key binding schemes with *t* enrollment observations, and it follows from the results presented in [24,25].

**Theorem** **1.**
*A secret-key rate $R_{t}$ is achievable after t enrollment observations if and only if*
(14)$$R_{t}\leq I\left(Y;X_{1},X_{2},\ldots ,X_{t}\right).$$
*The secret-key capacity $C_{t}=I\left(Y;X_{1},X_{2},\ldots ,X_{t}\right)$.*


In Figure 4, we plot the secret-key capacity for an example SRAM-PUF. Here, we have assumed a symmetric distribution of the one-probabilities, and average error probability $\bar{\psi }\approx 0.15$, see Figure 2 for the Θ-distribution. The mutual information increases from 0.26 for a single enrollment observation, to 0.50 after 20 enrollment observations. This corresponds to almost doubling the secret-key rate regarding single enrollment.

In the plot, we also visualize an upper bound to the secret-key capacity, given by
(15)$$\begin{array}{c} I\left(Y;X_{1},X_{2},\ldots ,X_{t}\right)\leq I\left(Y;X_{1},X_{2},\ldots ,X_{t},\Theta \right)=I\left(Y;\Theta \right)+I\left(Y;X_{1},X_{2},\ldots ,X_{t}|\Theta \right)\\ \overset{\left(a\right)}{=}I\left(Y;\Theta \right)=H\left(Y\right)-\int_{0}^{1}H\left(Y|\Theta =\theta \right)p_{\Theta }\left(\theta \right)\;d\theta , \end{array}$$
where $\left(a\right)$ follows from the Markov chain $Y↔\Theta ↔\left(X_{1},X_{2},\ldots ,X_{t}\right)$. From the weak law of large numbers, i.e., $\lim_{t\to \infty }\mathrm{Pr}\left(|\frac{1}{t}\sum_{i=1}^{t}x_{i}-E\left[X_{i}\right]|>\epsilon \right)=0$ for any ϵ>0, it follows that $\frac{1}{t}\sum_{i=1}^{t}x_{i}$ converges (in probability) to θ for t→∞. Therefore, the upper bound (15) can be achieved and we say
(16)$$C_{\infty }=I\left(Y;\Theta \right)=H\left(Y\right)-\int_{0}^{1}h_{2}\left(\theta \right)p_{\Theta }\left(\theta \right)\;d\theta .$$

## 4. Multiple-Observations Helper Data Scheme

We introduce the Multiple-Observations (MO) helper data scheme for binding a secret key to multiple observations of an SRAM-PUF, see Figure 5. First, we give a step-by-step description of the scheme. Then, we prove that it is secure. Finally, we prove that the scheme achieves the secret-key capacity $C_{t}$ (see Theorem 1) for *t* enrollment observations and, therefore, that the MO helper data scheme is optimal. Please note that all derivations are under the symmetric Θ-distribution assumption (Equation 6).

### 4.1. Description of the Scheme

First, a binary secret key $s\in {\left\{0,1\right\}}^{\log_{2}\left|S\right|}$ is generated uniformly at random, such that $H\left(S\right)=\log_{2}\left|S\right|$, with $\log_{2}\left|S\right|$ an integer. The key *s* is then encoded using a linear error-correcting code into a codeword $c^{n}$. The codeword is XORed with all *t* enrollment vectors $\left(x_{1}^{n},x_{2}^{n},\ldots ,x_{t}^{n}\right)$, resulting in *t* sequences $\left(w_{1}^{n},w_{2}^{n},\ldots ,w_{t}^{n}\right)$, with
(17)$$w_{i}^{n}=c^{n}⊕x_{i}^{n},\;\;\mathrm{for}\;i=1,2,\ldots ,t.$$
Please note that $w_{i}^{n}$ corresponds to a helper data sequence of the classic fuzzy commitment scheme [6]. The helper data of the MO helper data scheme is constructed by adding all these sequences together, i.e., the helper data are
(18)$$\begin{array}{ccc} m_{t}^{n} & \overset{\Delta }{=} & w_{1}^{n}+w_{2}^{n}+\ldots +w_{t}^{n}=\sum_{i=1}^{t}\left(c^{n}⊕x_{i}^{n}\right). \end{array}$$
The helper data are stored on the device or in a database, and they are considered public information.

When the key must be recovered, the decoder observes the helper data $m_{t}^{n}$ and another observation vector $y^{n}$ of the SRAM-PUF. A decoder function maps each pair $\left(m_{t}^{n},y^{n}\right)$ to a corresponding estimate $\hat{s}$ of the original secret. Ideally, the reconstructed secret $\hat{s}$ is equal to the original secret. Furthermore, an attacker who can observe the helper data $m_{t}^{n}$ should not obtain information about the secret.

### 4.2. Uniformity and Zero Leakage

First, by definition of the MO helper data scheme, the secret key is generated uniformly at random and therefore the uniformity condition of Definition 1, $\frac{1}{n}H\left(S\right)=\frac{1}{n}\log_{2}\left|S\right|$, is satisfied. Second, it has been shown in [17] that, for symmetric SRAM PUFs, zero leakage occurs by all the helper data after repeated enrollments, i.e., $I(S;W_{1}^{n},\ldots ,W_{t}^{n})=0$. It follows that
(19)$$\begin{array}{ccc} I\left(S;M_{t}^{n}\right) & \overset{\left(a\right)}{\leq } & I\left(C^{n};W_{1}^{n},W_{2}^{n},\ldots ,W_{t}^{n}\right)=0, \end{array}$$
where $\left(a\right)$ follows from the data processing inequality (see [19] Chapter 2) for the Markov Chain
(20)$$S↔C^{n}↔\left(W_{1}^{n},W_{2}^{n},\ldots ,W_{t}^{n}\right)↔M_{t}^{n}.$$
By (19) the MO helper data scheme achieves *strong secrecy*, i.e., $I\left(S;M_{t}^{n}\right)\leq \delta$.

### 4.3. Achievable Secret-Key Rate

Third, we derive the achievable rate over the channel from the encoder to the decoder. The channel is described by the conditional distribution $\mathrm{Pr}\left(M_{t}=m_{t},Y=y|C=c\right)$, and it follows from the channel coding theorem (see, e.g., [20], Chapter 3) that the maximum achievable rate is given by the mutual information maximized over the input distribution. Furthermore, *C* is uniform here, since we are using a linear code. We therefore evaluate this mutual information next
(21)$$\begin{array}{cc} R_{\mathrm{MO},t}^{*} & \overset{\Delta }{=}I\left(C;M_{t},Y\right)=I\left(C;M_{t}\right)+I\left(C;Y|M_{t}\right)\\ & \overset{\left(a\right)}{=}H\left(Y|M_{t}\right)-H\left(Y|M_{t},C\right)\\ & \overset{\left(b\right)}{=}H\left(Y|M_{t}\right)-H\left(Y|w_{H}\left(X_{1},X_{2},\ldots ,X_{t}\right),C\right)\\ & \overset{\left(c\right)}{=}1-H\left(Y|w_{H}\left(X_{1},X_{2},\ldots ,X_{t}\right)\right)\\ & \overset{\left(d\right)}{=}H\left(Y\right)-H\left(Y|X_{1},X_{2},\ldots ,X_{t}\right)\\ & =I\left(Y;X_{1},X_{2},\ldots ,X_{t}\right). \end{array}$$
Please note that $\left(a\right)$ follows from zero leakage (see (24)), and $\left(b\right)$ holds since (by (18)) the Hamming weight $w_{H}\left(x_{1},x_{2},\ldots ,x_{t}\right)$ is uniquely determined by $m_{t}$ given *c* and vice-versa. Furthermore, $\left(c\right)$ follows from (22) below and since the key and thus codebit *c* are generated independently from the SRAM-PUF observations. Finally, $\left(d\right)$ follows from the fact that the symmetric SRAM-PUF is unbiased (Equation 5), and by the fact that the conditional probability of the next observation given *t* previous observations $\left(x_{1},x_{2},\ldots ,x_{t}\right)$ only depends on the number of observed ones (7). The following derivation shows that the helper data integer $m_{t}$ by itself does not reveal any information about the reconstruction observation *y*, and thus $H(Y|M_{t})=1$.
(22)$$\begin{array}{ccc} \mathrm{Pr}\left(Y=1|M_{t}=m_{t}\right) & \overset{\left(a\right)}{=} & \sum_{c\in \{0,1\}}\mathrm{Pr}\left(C=c\right)\mathrm{Pr}\left(Y=1|M_{t}=m_{t},C=c\right)\\ & \overset{\left(b\right)}{=} & \begin{array}{c} \frac{1}{2}\mathrm{Pr}\left(Y=1|w_{H}\left(X_{1},\ldots ,X_{t}\right)=m_{t},C=0\right)\\ +\frac{1}{2}\mathrm{Pr}\left(Y=1|w_{H}\left(X_{1},\ldots ,X_{t}\right)=t-m_{t},C=1\right) \end{array}\\ & \overset{\left(c\right)}{=} & \begin{array}{c} \frac{1}{2}\mathrm{Pr}\left(Y=1|w_{H}\left(X_{1},\ldots ,X_{t}\right)=m_{t}\right)\\ +\frac{1}{2}\mathrm{Pr}\left(Y=0|w_{H}\left(X_{1},\ldots ,X_{t}\right)=m_{t}\right)=\frac{1}{2}, \end{array} \end{array}$$
where $\left(a\right)$ follows from (23), $\left(b\right)$ follows from the definition of the helper data (18), and in $\left(c\right)$ we use the fact that the key and codebit *c* is generated independently from the SRAM-PUF observations, and furthermore we use the symmetry property for the conditional distribution (8) that we have derived for symmetric SRAM-PUF in Section 2. In the above derivation we use that
(23)$$\mathrm{Pr}\left(C=c|M_{t}=m_{t}\right)=\mathrm{Pr}\left(C=c\right),$$
which can be derived from (19) by observing that
(24)$$I\left(C;M_{t}\right)\overset{\left(a\right)}{\leq }I\left(S;M_{t}^{n}\right)=0,$$
where $\left(a\right)$ follows from the data processing inequality (see [19], Chapter 2) for the Markov Chain
(25)$$C↔C^{n}↔S↔M_{t}^{n}↔M_{t}.$$

It follows from the derivations in Section 4.2 and Section 4.3 that the MO helper data scheme achieves secret-key rate $R_{\mathrm{MO},t}^{*}$, where achievable secret-key rate is defined in Definition 1. Furthermore, $R_{\mathrm{MO},t}^{*}=C_{t}$ is equal to the secret-key capacity for *t* enrollment observations as given by Theorem 1, which shows that the MO helper data scheme is optimal.

## 5. Code Construction and Simulation Results

In the previous section, we have proved that secret-key capacity for *t* enrollment observations is achievable with the MO helper data scheme. However, the performance that is achieved in practice depends on the error-correcting code that is implemented by the encoder and decoder function. Here, performance is evaluated in terms of secret-key rate (*R*) and reconstruction error probability (FER). Until now, we have not specified an error-correcting code. Please note that a Soft-Decision decoder should be used that can benefit from the reliability information of the SRAM cells. Furthermore, we are looking for codes that have a low rate and that perform well for relatively short blocklengths (128 key bits). Here, we use an off-the-shelf LDPC code to evaluate the expected performance of the scheme in a practical setting.

### 5.1. Encoder

First, a key *s* of 128 bits length is generated uniformly at random. The secret-key is then encoded using an error-correcting code, resulting in codeword $c^{n}$. As the error-correcting code we use a CRC (cyclic-redundancy check) code concatenated with an LDPC code as defined in the 5G NR (fifth generation new radio) standard [26]. For our simulations, we use the LDPC coding-chain implementation from the MATLAB 5G Toolbox [27]. The codeword $c^{n}$ is XORed with *t* SRAM-PUF observation vectors of length *n* and the results are added together, resulting in helper data $m_{t}^{n}$. This concludes the encoder part of the scheme, where the helper data $m_{t}^{n}$ is stored for later usage and all other variables are discarded.

The secret-key rate of our implementation is
(26)$$R=\frac{128}{n}\;\mathrm{key}\;\mathrm{bits}\;\mathrm{per}\;\mathrm{SRAM}\;\mathrm{cell}.$$
Furthermore, for *t* enrollment observations of *n* SRAM cells the required number of bits for storage of the helper data are $n\lceil \log_{2}\left(t+1\right)\rceil$, and thus the helper data rate is
(27)$$R^{\mathrm{hd}}=\frac{\left\lceil \log_{2}\left(t+1\right)\right\rceil }{R}\;\mathrm{helper}\;\mathrm{data}\;\mathrm{bits}\;\mathrm{per}\;\mathrm{key}\;\mathrm{bit}.$$

### 5.2. Decoder

The decoder uses an SRAM-PUF observation $y^{n}$ and the previously stored helper data $m_{t}^{n}$ to reconstruct the key $\hat{s}$. First, the log-likelihood ratios (LLR) of the received code bits are calculated. Since the SRAM cells are independently distributed, we can calculate the LLR of each bit separately. The LLR of a code bit *c*, after observing the corresponding helper data $m_{t}$ and SRAM-PUF observation *y*, is
(28)$$\mathrm{LLR}\left(y,m_{t}\right)=\log\frac{\pi_{t+1}\left(y+m_{t}\right)}{\pi_{t+1}\left(y+t-m_{t}\right)}.$$
See the Section A.1 for a derivation of the above equation. Please note that there are $2\left(t+1\right)$ possible combinations of $\left(y,m_{t}\right)$. However, due to the symmetry properties of the LLR function (see Section A.1), we only need to store $\frac{\left(t+2\right)}{2}$ LLRs in a look-up table. Finally, the LDPC decoder uses an iterative Soft-Decision decoder (belief propagation algorithm), combined with CRC for error detection, to reconstruct the 128 bit secret $\hat{s}$ from the received LLRs.

### 5.3. Simulation Results

We have simulated the MO helper data scheme using Monte Carlo simulations and the statistical model for SRAM PUFs that was presented in Section 2, with $\lambda_{1}=0.51$ and $\lambda_{2}=0$ and average error probability $\bar{\psi }\approx 0.15$. We plot the resulting error probability of the key reconstruction (FER) for various key rates *R* and number of enrollment observations *t* in Figure 6.

Observe that the error probability can be reduced by decreasing the secret-key rate *R* and by increasing the number of enrollment observations *t*. We choose $\mathrm{FER}\leq 10^{-6}$ as the target error probability and find that R=1/6 is the best (highest) achievable secret-key rate that achieves the target FER. Furthermore, at least 5 enrollment observations are required to achieve the target FER with this key rate, and the corresponding helper data rate is $R^{\mathrm{hd}}\approx 15.5$. Another rate-enrollment pair that achieves the target FER is R=1/7 for t=3 enrollment observations, with corresponding helper data rate $R^{\mathrm{hd}}\approx 14$. Please note that the classic single enrollment scheme (corresponding to MO scheme for t=1) achieves the target FER for R=1/11 or worse key rates. Furthermore, the helper data rate $R^{\mathrm{hd}}=11$ bits per key bit in this case. Therefore, the MO helper data scheme achieves a secret-key rate that is 11/6≈1.8 times higher than the single enrollment scheme. However, the improved secret-key rate comes at a cost since the helper data rate is $\frac{15.5}{11}\approx 1.4$ times as high in comparison to the classic scheme.

We conclude that a trade-off must be considered between the required enrollment time, helper data storage, and the number of SRAM cells, when selecting the parameters (enrollment observations and secret-key rate) for a given setting.

## 6. The Soft-Decision Helper Data Scheme

The performance increase of the MO helper data scheme, regarding the traditional single enrollment scheme, is mostly due to the fact that it can distinguish between reliable and unreliable SRAM cells. A well-known helper data scheme that also considers the reliability information of the SRAM cells is the Soft-Decision scheme introduced by Maes et al. [7]. In this section, we first describe the SD scheme and derive its achievable performance. We note that the SD scheme assumes that the one-probabilities of the SRAM cells are observable which in practice is often not the case. Therefore, a pre-processing step is required that estimates the one-probabilities of the SRAM cells. Based on this observation, we propose a variation on the SD scheme that instead directly considers the binary SRAM-PUF observations as an input. We show that the newly proposed strategy results in the same decoder LLRs as the MO scheme (for equal key and SRAM observations). This implies that both schemes achieve the same reconstruction performance, and thus, since the MO scheme is provably optimal, the newly proposed strategy for the binary SD scheme is optimal as well.

To the best of our knowledge, we are the first to propose an optimal strategy for the SD scheme with binary enrollment observations. Furthermore, we show that very few observations (less than 10) are sufficient to achieve an acceptable performance, whereas in the literature 64 observations are used [8] for the same statistical model and parameter settings that are used in the current work.

### 6.1. Description of (Regular) SD Helper Data Scheme

The SD helper data scheme, see Figure 7, observes the one-probability vector $\theta^{n}$ of all the cells, and derives the dominant values $u^{n}$, defined as
(29)$$u\overset{\Delta }{=}\left\{\begin{array}{cc} 0 & \mathrm{if}\;0\leq \theta \leq 1/2,\\ 1 & \mathrm{if}\;1/2<\theta \leq 1, \end{array}\right.$$
and the error probability $\psi^{n}$ (regarding the dominant value of the cell), defined as
(30)$$\psi \overset{\Delta }{=}\min\left(\theta ,\left(1-\theta \right)\right)=\left\{\begin{array}{cc} \theta & \mathrm{if}\;0\leq \theta \leq 1/2,\\ 1-\theta & \mathrm{if}\;1/2<\theta \leq 1, \end{array}\right.$$
with ψ∈[0,1/2]. As in the MO helper data scheme, a key is generated uniformly at random, and encoded using a linear error-correcting code into a codeword $c^{n}$. Then the codeword is XORed with the dominant values $u^{n}$ to create the helper data sequence
(31)$$w^{n}=c^{n}⊕u^{n}.$$

Besides the helper data sequence $w^{n}$, also the reliability values $\psi^{n}$ of the SRAM cells are stored publicly. For reconstruction, another observation $y^{n}$ of the SRAM-PUF is XORed with the helper data, resulting in a noisy codeword
(32)$$r^{n}=w^{n}⊕y^{n}=c^{n}⊕\left(u^{n}⊕y^{n}\right),$$
where $\left(u^{n}⊕y^{n}\right)$ is an error vector, and an error ($r_{j}=c_{j}⊕1$) occurs when the reconstruction observation $y_{j}$ is flipped regarding the dominant value $u_{j}$ of a cell. The decoder reconstructs the key $\hat{s}$ based on the received noisy codeword $r^{n}$ and the error probabilities of the cells $\psi^{n}$.

### 6.2. Achievable Performance

We are interested in the maximum achievable secret-key rate of the SD scheme. As for the MO scheme (Section 4.2) we should first show that the SD scheme is secure, i.e., the uniformity condition of Definition 1 is satisfied since the key is generated uniformly at random. Furthermore, the leakage condition is satisfied as $I(S;W^{n},\Psi^{n})=0$ as is shown in Section A.2. Now, we can derive the achievable rate over the channel from the encoder to the decoder. The channel is described by the conditional distribution $\mathrm{Pr}\left(R=r,\Psi =\psi |C=c\right)$, and it follows from the channel coding theorem (see, e.g., [20], Chapter 3) that the maximum achievable rate is given by the mutual information maximized over the input distribution. Furthermore, *C* is uniform here, since we are using a linear code. We therefore evaluate this mutual information next
(33)$$\begin{array}{ccc} R_{\mathrm{SD},\infty }^{*} & \overset{\Delta }{=} & I\left(C;R,\Psi \right)\overset{\left(a\right)}{=}I\left(C;R|\Psi \right)\overset{\left(b\right)}{=}1-\int_{0}^{1/2}h_{2}\left(\psi \right)p_{\Psi }\left(\psi \right)\;d\psi . \end{array}$$
In (a) we used the fact that the key is generated independently from the SRAM-PUF observations and thus $I\left(C;\Psi \right)=0$. In (b) we used the fact that r=c⊕(u⊕y) so the channel from encoder to decoder can be modeled as a binary symmetric channel with cross-over probability ψ (the probability that u≠y). Please note that $R_{\mathrm{SD},\infty }^{*}=C_{\infty }$ (for symmetric SRAM-PUF) this rate is equal to the limit (for *t* to infinity) of the secret-key capacity for multiple enrollment observations, see (Equation 10). Therefore, the SD helper data scheme is optimal, and furthermore the MO helper data scheme approaches the same key rate as the SD helper data scheme when sufficient enrollment observations are used.

### 6.3. New SD Strategy for Binary Enrollment Observations

In practice, the one-probabilities $\theta^{n}$ are non-observable and therefore, they must be estimated based on the binary observation vectors $\left(x_{1}^{n},x_{2}^{n},\ldots ,x_{t}^{n}\right)$ before the SD scheme can be applied. Instead of applying a pre-processing step for estimation, we propose to adjust the SD scheme s.t. the dominant values and error probabilities are directly estimated based on *t* binary observations. The dominant value of an SRAM cell is then estimated as
(34)$${\hat{u}}_{t}\overset{\Delta }{=}\left\{\begin{array}{cc} 0 & \mathrm{if}\;w_{H}\left(x_{1},\ldots ,x_{t}\right)\leq t/2,\\ 1 & \mathrm{otherwise}. \end{array}\right.$$
And furthermore, the error probability (regarding the dominant value) is
(35)$${\hat{\psi }}_{t}\overset{\Delta }{=}\frac{\pi_{t+1}\left(1+m_{t}^{\mathrm{SD}}\right)}{\pi_{t}\left(m_{t}^{\mathrm{SD}}\right)},$$
with
(36)$$\begin{array}{ccc} m_{t}^{\mathrm{SD}} & \overset{\Delta }{=} & \min\left(w_{H}\left(x_{1},\ldots ,x_{t}\right),t-w_{H}\left(x_{1},\ldots ,x_{t}\right)\right)\\ & = & \left\{\begin{array}{cc} w_{H}\left(x_{1},\ldots ,x_{t}\right) & \mathrm{if}\;w_{H}\left(x_{1},\ldots ,x_{t}\right)\leq t/2,\\ t-w_{H}\left(x_{1},\ldots ,x_{t}\right) & \mathrm{otherwise}. \end{array}\right. \end{array}$$
In Section A.3 we show that (36) indeed is equal to the error probability (regarding the dominant value) of a cell given the *t* binary enrollment observations.

The helper data and noisy codeword are constructed as before, so $w^{n}=c^{n}⊕{\hat{u}}_{t}^{n}$ and $r^{n}=c^{n}⊕\left({\hat{u}}_{t}^{n}⊕y^{n}\right)$. The decoder reconstructs the key $\hat{s}$ based on the received noisy codeword $r^{n}$ and the error probabilities of the cells ${\hat{\psi }}_{t}^{n}$.

### 6.4. Achievable Performance

We are interested in the maximum achievable secret-key rate of the binary SD scheme. As before we should first show that the scheme is secure, i.e., the uniformity condition of Definition 1 is satisfied since the key is generated uniformly at random. Furthermore, the leakage condition is satisfied as $I(S;W^{n},{\hat{\Psi }}_{t}^{n})=0$ as is shown in Section A.4. Now, we can derive the achievable rate over the channel from the encoder to the decoder. The channel is described by the conditional distribution $\mathrm{Pr}\left(R=r,{\hat{\Psi }}_{t}={\hat{\psi }}_{t}|C=c\right)$, and it follows from the channel coding theorem (see, e.g., [20], Chapter 3) that the maximum achievable rate is given by the mutual information maximized over the input distribution. Furthermore, *C* is uniform here, since we are using a linear code. We therefore evaluate this mutual information next
(37)$$\begin{array}{ccc} R_{\mathrm{SD},t}^{*} & \overset{\Delta }{=} & I\left(C;R,{\hat{\Psi }}_{t}\right)\overset{\left(a\right)}{=}I\left(C;R|{\hat{\Psi }}_{t}\right)=H\left(R|{\hat{\Psi }}_{t}\right)-H\left(R|{\hat{\Psi }}_{t},C\right)\\ & \overset{\left(b\right)}{=} & 1-H\left({\hat{U}}_{t}⊕Y|{\hat{\Psi }}_{t}\right)\overset{\left(c\right)}{=}1-H\left({\hat{U}}_{t}⊕Y|M_{t}^{\mathrm{SD}},{\hat{U}}_{t}\right)=1-H\left(Y|M_{t}^{\mathrm{SD}},{\hat{U}}_{t}\right)\\ & \overset{\left(d\right)}{=} & 1-H\left(Y|w_{H}\left(X_{1},X_{2},\ldots ,X_{t}\right)\right)\overset{\left(e\right)}{=}H\left(Y\right)-H\left(Y|X_{1},X_{2},\ldots ,X_{t}\right)\\ & = & I\left(Y;X_{1},X_{2},\ldots ,X_{t}\right). \end{array}$$
In (a) we used the fact that the key is generated independently from the SRAM-PUF observations and thus $I\left(C;{\hat{\Psi }}_{t}\right)=0$. In $\left(b\right)$ we used that $r=c⊕({\hat{u}}_{t}⊕y)$ and furthermore the key (and codebit) is generated uniformly and independently from the SRAM-PUF observations (and thus also independently from ${\hat{\psi }}_{t}$). Furthermore, $\left(c\right)$ follows by the Markov Chain
(38)$$Y⊕{\hat{U}}_{t}↔{\hat{\Psi }}_{t}↔\left(M_{t}^{\mathrm{SD}},{\hat{U}}_{t}\right),$$
and $\left(d\right)$ holds since (by (18)) $w_{H}\left(x_{1},x_{2},\ldots ,x_{t}\right)$ is uniquely determined by the tuple $\left(m_{t}^{\mathrm{SD}},{\hat{u}}_{t}\right)$ and vice-versa. Finally, $\left(e\right)$ follows from the fact that the symmetric SRAM-PUF is unbiased (Equation 5), and by the fact that the conditional probability of the next observation given *t* previous observations $\left(x_{1},x_{2},\ldots ,x_{t}\right)$ only depends on the number of observed ones (7).

It follows from the derivations above that the binary SD scheme achieves secret-key rate $R_{\mathrm{SD},t}^{*}$, where achievable secret-key rate is defined in Definition 1. Furthermore, $R_{\mathrm{SD},t}^{*}=C_{t}$ is equal to the secret-key capacity for *t* enrollment observations as given by Theorem 1, which shows that the binary SD scheme is optimal.

### 6.5. Code Construction and Simulations

As with the MO scheme we can now define a code construction, with a uniformly generated secret *s*, an encoder based on an off-the-shelf LDPC code, and where the helper data $w^{n}$ and reliability information ${\hat{\psi }}_{t}^{n}$ are constructed as explained in Section 6.3.

The decoder observes $r^{n}$ and the error probabilities ${\hat{\psi }}_{t}$ which can be used to calculate the log-likelihood ratios needed for reconstruction of the secret. Since the SRAM cells are independently distributed, we can calculate the LLR of each bit separately. The LLR for a codebit *c*, after observing the noisy codebit *r* and the error probability ${\hat{\psi }}_{t}$ for *t* enrollment observations, is
(39)$$\begin{array}{ccc} {\mathrm{LLR}}^{\mathrm{SD}}\left(r,{\hat{\psi }}_{t}\right) & = & \left\{\begin{array}{cc} \log\frac{{\hat{\psi }}_{t}}{1-{\hat{\psi }}_{t}} & \mathrm{if}\;r=1,\\ \log\frac{1-{\hat{\psi }}_{t}}{{\hat{\psi }}_{t}} & \mathrm{otherwise}. \end{array}\right. \end{array}$$
See Section A.5 for a derivation.

It is shown in Section A.5 that the LLRs for the SD scheme are equal to the LLRs for the MO scheme when the same enrollment observations $\left(x_{1},\ldots ,x_{t}\right)$ and reconstruction observation *y* are generated by the SRAM-PUF. Therefore, simulations for the SD scheme would give the same FER results as for the MO scheme, and we can use the plots in Figure 6 to predict the performance of the SD scheme.

## 7. Sequential MO Helper Data Scheme

We have seen in Section 5 that a better performance (smaller reconstruction error probability) is often achieved when the number of enrollment observations is increased. However, each enrollment observation requires a full reset (power off and power up) of the SRAM-PUF. Therefore, considering more enrollment observations results in an increased duration of the enrollment phase of the MO helper data scheme. This may be undesirable in practice as the enrollment phase must be completed in a secure environment and limited time is available.

In this section, we present a variation on the MO helper data scheme that we call the Sequential Multiple-Observations (SMO) helper data scheme. We exploit the fact that a new SRAM-PUF observation is required for each key reconstruction, and, after the reconstruction, we use this additional observation to update the helper data. As a result, the helper data are updated (sequentially) after each reconstruction, and thus the reconstruction error rate is reduced over time. The SMO scheme improves the efficiency regarding the MO scheme, since observations are used both for reconstruction and enrollment. Furthermore, the SMO scheme enables the usage of a reduced number of observations during the enrollment phase. The reduced number of enrollment observations comes at an initial cost of larger reconstruction error probability (FER); however, this is quickly reduced (see Figure 8) during the lifetime of the device.

### 7.1. Description

The enrollment phase of the Sequential MO helper data scheme is the same as for the MO helper data scheme, i.e., a key *s* is generated uniformly at random, and the helper data are constructed according to (18), with $t=t_{0}$ enrollment observations. Therefore, after the initial enrollment, a helper data $m_{t_{0}}^{n}$ is stored that is based on $t_{0}$ enrollment observations. The reconstruction phase, however, is different than before and it is visualized in Figure 8.

In the reconstruction phase of the SMO scheme, the helper data $m_{t}^{n}$ (corresponding to *t* SRAM-PUF observations) and an SRAM-PUF observation $y^{n}$ are used to reconstruct the key $\hat{s}$. If reconstruction is successful (We assume that correctness of the key can be verified, for example, by feedback from other protocols that apply the key for decryption, or by a failed authentication.), the SRAM-PUF observation is XORed with the encoded secret and is added to the existing helper data, which results in a new (improved) helper data corresponding to t+1 SRAM-PUF observations. If, on the other hand, reconstruction has failed, it is not possible to reconstruct the original codeword $c^{n}$ which is required for the helper data update. Therefore, the helper data remains the same as before in this case.

### 7.2. Security and Achievable Rate

The main difference between the Sequential MO helper data scheme and the original MO helper data scheme is that the helper data are now updated sequentially in the field. Therefore, an attacker may have access to multiple helper data $\left(m_{1}^{n},m_{2}^{n},\ldots ,m_{t}^{n}\right)$, instead of only the final helper data $m_{t}^{n}$. However, despite the reveal of multiple helper data, we can show that zero-secrecy leakage is still guaranteed. In particular, a similar Markov Chain as defined in (20) holds in this case
(40)$$S↔C^{n}↔\left(W_{1}^{n},W_{2}^{n},\ldots ,W_{t}^{n}\right)↔\left(M_{1}^{n},M_{2}^{n},\ldots ,M_{t}^{n}\right).$$
Therefore, we can repeat the same derivations as in Section 4.2 to show that
(41)$$I\left(S;M_{1}^{n},M_{2}^{n},\ldots ,M_{t}^{n}\right)\leq I\left(C^{n};W_{1}^{n},W_{2}^{n},\ldots ,W_{t}^{n}\right)=0,$$
and thus, there is no information leakage about the key by all the revealed helper data.

Please note that only the helper data are updated, whereas the key that has been generated during the enrollment phase remains the same. Since the key should be reconstructable already after the initial enrollment (since successful reconstruction is required for each update), the maximum achievable rate of the Sequential MO scheme is limited by the number of observations $t_{0}$ that is used for the initial enrollment, i.e.,
(42)$$R_{\mathrm{SMO},t_{0}}^{*}=R_{\mathrm{MO},t_{0}}^{*}=I\left(Y;X_{1},X_{2},\ldots ,X_{t_{0}}\right).$$
Nevertheless, as we show in the next subsection, the main advantage of the SMO scheme is that the reconstruction observations are exploited to improve the reconstruction error probability over time.

### 7.3. Simulation Results

We evaluate the performance of the Sequential MO helper data scheme through Monte Carlo simulations. The encoder and decoder constructions and LLR calculations are similar to the construction presented in Section 5. We evaluate two key rates, R=1/6 and R=1/7 with $t_{0}=4$ and $t_{0}=2$ enrollment observations, respectively. We have chosen the number of enrollment observations s.t. the achieved initial error probability is larger than, but close to, the target $\mathrm{FER}<10^{-6}$ (based on the previous results in Figure 6). We simulate 4 consecutive reconstruction attempts of the SMO scheme and plot the FER as function of the number of reconstruction attempts (time) in Figure 9. Please note that the helper data are only updated when the reconstruction attempt is successful. Therefore, the ’reconstructions’ in the current simulation are different from ’*t* enrollment observations’ plotted on the x-axis in Figure 9.

First, we can directly observe that the FER reduces with each reconstruction attempt. Therefore, indeed the SMO scheme supports an improved reconstruction performance over the lifetime of the devices. More specifically, for the simulated rates and selected number of enrollment observations, we see that the average error probability already reduces below the required threshold $\mathrm{FER}<10^{-6}$ after the first reconstruction attempt. Therefore, in both cases, the initial cost (larger FER) of reducing the number of enrollment observations by one, was already nullified after one reconstruction.

## 8. Conclusions

We have presented the Multiple-Observations (MO) helper data scheme for binding a secret key to multiple observations of an SRAM-PUF. We have shown that the MO helper data scheme can achieve secret-key capacity corresponding to *t* enrollment observations, and therefore the scheme is optimal in information-theoretic sense. Furthermore, we have evaluated performance of the scheme with Monte Carlo simulations for a standard statistical model for SRAM PUFs with average error probability $\bar{\psi }\approx 0.15$. Secret-key rate R=1/6 is sufficient to achieve $\mathrm{FER}\leq 10^{-6}$ after t=5 enrollment observations. This is a key rate that is 11/6≈1.8 times higher (better) than for the single enrollment scheme with comparable FER.

The MO helper data scheme is very similar to the Soft-Decision (SD) helper data scheme; however, the SD scheme assumes one-probabilities as an input, which in practice are non-observable. Therefore, we proposed a new strategy that considers binary observations instead. We have shown that this new strategy is optimal and achieves the same reconstruction performance as the MO scheme.

We have introduced a variation on the MO scheme, which we call the Sequential Multiple-Observations helper data scheme. The scheme supports a sequential update of the helper data after each successful reconstruction of the key, resulting in a reduced FER over the lifetime of the device. The SMO scheme enables the usage of less enrollment observations, by accepting a (slightly) worse initial FER that is quickly improved upon.

## Figures and Tables

**Figure 1 entropy-23-00590-f001:**
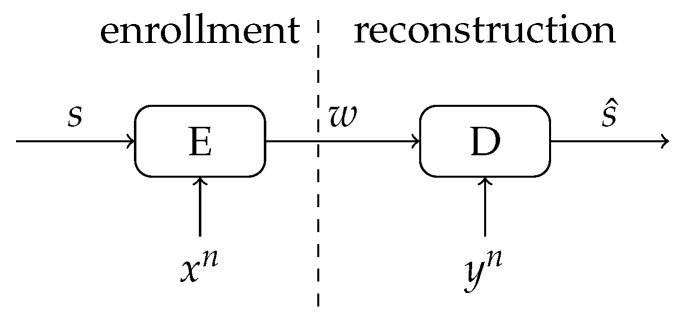
Key binding scheme for reliable secret-key reconstruction from noisy observations.

**Figure 2 entropy-23-00590-f002:**
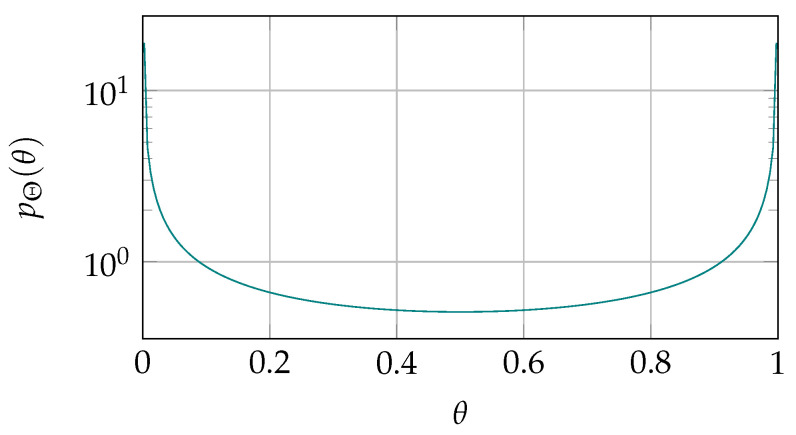
Distribution of the one-probability Θ as used for the simulations in this work. Generated according to the statistical model for SRAM-PUF [7] with $\lambda_{1}=0.51$, and $\lambda_{2}=0$.

**Figure 3 entropy-23-00590-f003:**
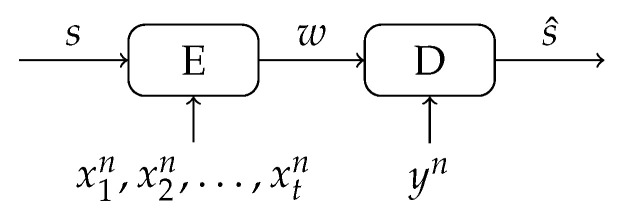
Secret-key binding with *t* observations by the encoder and a single observation by the decoder.

**Figure 4 entropy-23-00590-f004:**
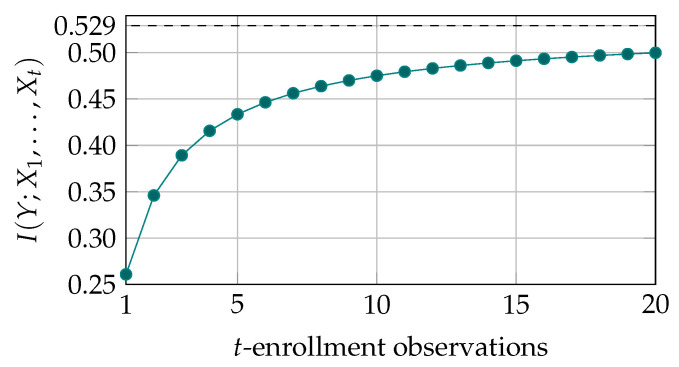
Secret-key capacity in bits per SRAM cell for the secret-key binding scheme with *t* enrollment observations, evaluated for SRAM PUFs given the Θ-distribution shown in Figure 2. The curve is approaching a limit that is represented by the dashed horizontal line.

**Figure 5 entropy-23-00590-f005:**
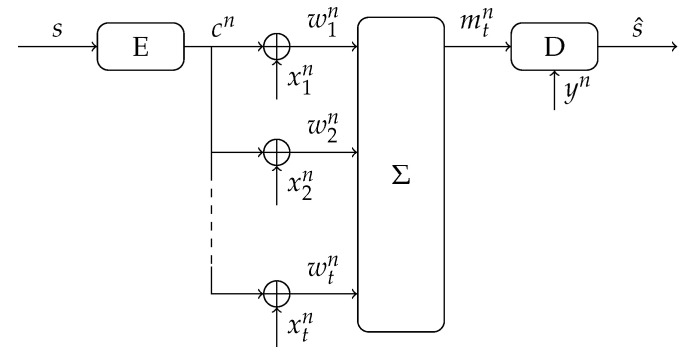
MO helper data scheme: *t* observations of the SRAM-PUF are used to generate a helper data $m_{t}^{n}$ that is published and that can be used by the decoder (together with an additional SRAM-PUF observation) to reconstruct the key.

**Figure 6 entropy-23-00590-f006:**
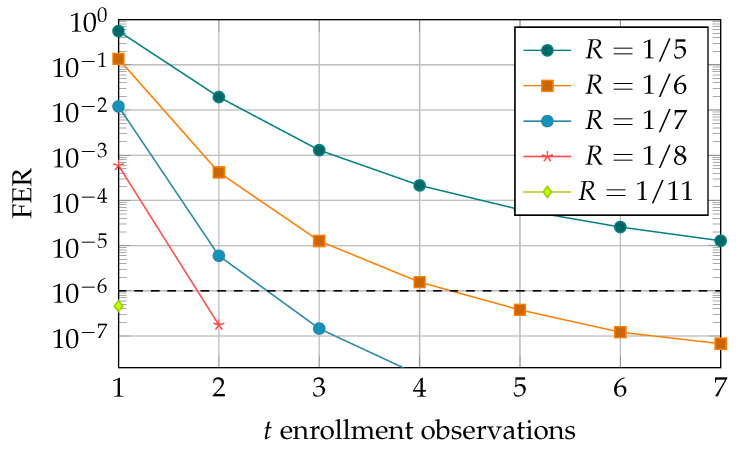
Reconstruction error probability FER for the MO helper data scheme, with 128 bit key and 128/R SRAM cells.

**Figure 7 entropy-23-00590-f007:**
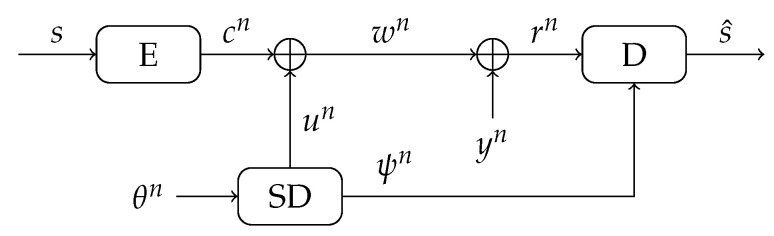
The Soft-Decision helper data scheme, with public information the helper data sequence $w^{n}$ and error probabilities $\psi^{n}$.

**Figure 8 entropy-23-00590-f008:**
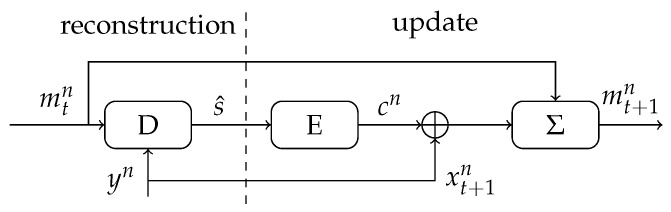
Reconstruction phase of the Iterative MO helper data scheme: the helper data $m_{t}^{n}$ is updated to $m_{t+1}^{n}$ after each successful reconstruction.

**Figure 9 entropy-23-00590-f009:**
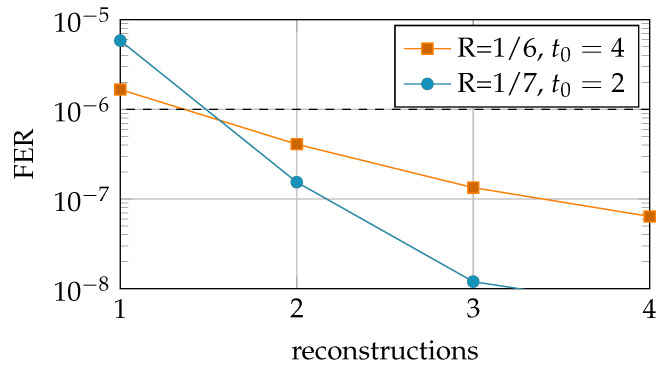
Reconstruction error probability FER for consecutive reconstructions for the Sequential MO helper data scheme, with 128 bit key.

## Data Availability

The plots in Figure 2, Figure 4 Figure 6 and Figure 9 are based on calculations and simulations performed in MATLAB. The scripts can be found at https://github.com/TUe-ICTLab/Multiple-Observations-for-Secret-Key-Binding-with-SRAM-PUFs (accessed on 7 April 2021).

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
