# Peer review of "Multiple Observations for Secret-Key Binding with SRAM PUFs"

_entropy, 2021, doi:10.3390/e23050590_

Round 1

Reviewer 1 Report

This is a nice work that combines theoretical results in terms fundamental limits (secrecy capacity), with practical simulations using realistic channel codes. It is shown that multiple observations can improve both capacity and practical performance. The connections made between the MO scheme and SD schemes are also a nice contribution. Based on the proposed techniques, the authors are able to improve upon several results in the literature.

The paper is very well written.

I have one question that the authors perhaps can easily answer: is there a way that the attacker can take advantage of a stronger decoder for the LDPC code? Or is the secrecy unaffected by a coding gain that a better decoder could obtain? Perhaps this could be commented somewhere if it is relevant. The reason for my comment is that LDPC codes perform far from optimal for length 128, compared to e.g. an ML decoder. And additionally, an overall ML decoder over LDPC and CRC together can outperform it further. Techniques like OSD could be easily applied to the joint CRC/LDPC codes.

Author Response

We thank the reviewer for taking the time to read our paper and for the comments and feedback.

The reviewer asked the following question:

I have one question that the authors perhaps can easily answer: is there a way that the attacker can take advantage of a stronger decoder for the LDPC code? Or is the secrecy unaffected by a coding gain that a better decoder could obtain? Perhaps this could be commented somewhere if it is relevant. The reason for my comment is that LDPC codes perform far from optimal for length 128, compared to e.g. an ML decoder. And additionally, an overall ML decoder over LDPC and CRC together can outperform it further. Techniques like OSD could be easily applied to the joint CRC/LDPC codes.

Our answer is as follows:

No, the attacker cannot take advantage of a stronger decoder. The reason for this is that the attacker can only access the helper data (m_t^n) which does not reveal any information about the codeword. This follows from equation (19) in the paper, which states that the mutual information I(M_t^n ; S) = 0, and thus the helper data M_t^n is independently distributed from the secret S (and the corresponding codeword C^n). Therefore, from the attacker’s perspective each possible codeword is equally likely, and even the best decoding techniques (e.g., ML) would not help the attacker to learn anything.

We hope this answers your question.

Reviewer 2 Report

This paper presented a new multiple-observations helper data scheme for secret key binding to an SRAM PUF. This approach binds a single key to multiple enrollment observations of the SRAM PUF. The scheme is evaluated through Monte Carlo simulations and the performance is significantly improved. The contribution is clearly described and descriptions are in detail. For this reason, I have some minor comments on this paper.

- The author utilized Soft-Decision designed for binary observation to the multiple-observation. I wonder there is the scheme firstly designed for multiple-observation or not.

- This paper utilized the LDPC code to correct errors in the data. There are many code algorithms such as Quasi-cyclic (QC) LDPC code. Please provide the opinion that why LDPC is optimal in this scheme.

- I want to see the performance comparison between multiple single-observation based PUF and multiple-observation based PUF.

- The evaluation is not clearly done with previous schemes. Please provide the comparison table with previous works.

Author Response

We thank the reviewer for taking the time to read our paper and for the comments and feedback. 

In the following, we offer a point-by-point response to the reviewers' detailed comments.

1) The author utilized Soft-Decision designed for binary observation to the multiple-observation. I wonder there is the scheme firstly designed for multiple-observation or not.

Our response:

We have initially developed the scheme for utilizing multiple observations for improving the decoder performance. Later, we noticed a close similarity to the existing Soft-Decision (SD) scheme introduced by Maes et al. [7]. Although the original SD scheme assumes direct observation of the soft (reliability) information, in practice the reliability information should be estimated based on multiple binary observations. In our work we introduce a new variation on the SD scheme that does not require a separate step for estimation of the reliability values. Then we show that this variation of the SD scheme is actually very similar to the MO scheme and that both schemes achieve the same performance in terms of error-probability and secret-key rate.

2) This paper utilized the LDPC code to correct errors in the data. There are many code algorithms such as Quasi-cyclic (QC) LDPC code. Please provide the opinion that why LDPC is optimal in this scheme.

Our response:

In the paper we show that the MO scheme is information-theoretically optimal and we derive maximum achievable rates. However, this does not tell us how well the scheme will perform in practice, especially since often relatively short error-correcting codes are used for the SRAM-PUF schemes, whereas capacity (maximum rate) is often achieved only for longer codewords. Therefore, we evaluate the performance of our scheme in a practical setting as well in Section 5. As explained in the paper, we need to apply an error-correcting code that uses a soft-decision decoder, such that it can benefit from the reliability information extracted from the multiple observations. An example of such a code is an LDPC code. We used an LDPC+CRC code from the 5G standard which is also included in the 5G toolbox for MATLAB. This makes it easy for the reader to reproduce and verify our results. Note that we do not claim that the used LDPC+CRC code is an optimal error-correcting code for this scenario.

3) I want to see the performance comparison between multiple single-observation based PUF and multiple-observation based PUF.

Our response:

The focus of this paper is to show how multiple observations can be exploited to improve the performance of the key binding schemes. To this end we are not focussing on using specific error-correcting codes, but only on evaluating the improved performance of the scheme as function of the used number of observations. We also compare the performance of the multiple observations scheme to a single observation scheme that uses exactly the same error-correcting code (LDPC+CRC). See Section 5.3. We show that using multiple observations we can improve the secret-key rate from 1/11 (single enrollment) to 1/6 (5 enrollments).

4) The evaluation is not clearly done with previous schemes. Please provide the comparison table with previous works.

Our response:

We cannot provide a fair comparison w.r.t. other schemes, as different error-correcting codes are used for each scheme, whereas error-correcting codes are not the focus of this paper. Furthermore, the SD scheme and SI schemes from the literature assume a different input (reliability information of the SRAM cells) than the MO scheme (binary observations of the SRAM cells). Therefore, we instead focused on studying the similarity between the MO scheme and the existing SD scheme [7] in Section 6. In Section 6 we introduce a variant on the SD scheme that considers binary values as an input. We show that this new SD scheme is in principle equivalent to the MO scheme, and that both schemes achieve the same performance. If instead we would have compared the MO scheme to the results presented for the SD scheme in the literature, we would have found that MO is superior to SD, since literature [8] reports a key rate of 1/12 for SD and we have found rate 1/6 for MO. This difference is due to the fact that different error-correcting codes were used.